# Religious Affiliation in Relation to Positive Mental Health and Mental Disorders in a Multi-Ethnic Asian Population

**DOI:** 10.3390/ijerph18073368

**Published:** 2021-03-24

**Authors:** Janhavi Ajit Vaingankar, Niyanta Choudhary, Siow Ann Chong, Fiona Devi Siva Kumar, Edimansyah Abdin, Saleha Shafie, Boon Yiang Chua, Rob M. van Dam, Mythily Subramaniam

**Affiliations:** 1Institute of Mental Health, Research Division, Singapore 539747, Singapore; niyanta2000@gmail.com (N.C.); siow_ann_chong@imh.com.sg (S.A.C.); Fiona_Devi_SIVA_KUMAR@imh.com.sg (F.D.S.K.); edimansyah_abdin@imh.com.sg (E.A.); saleha_shafie@imh.com.sg (S.S.); boon_yiang_chua@imh.com.sg (B.Y.C.); mythily@imh.com.sg (M.S.); 2Saw Swee Hock School of Public Health, Yong Loo Lin School of Medicine, National University of Singapore, Singapore 117597, Singapore; rob.van.dam@nus.edu.sg; 3National University Health System, Singapore 119228, Singapore

**Keywords:** Christianity, Composite International Diagnostic Interview, eastern religions, emotional support, interpersonal skills, personal growth and autonomy, spirituality

## Abstract

Background: This study investigated association of religious affiliation with positive mental health (PMH) and mental disorders. Methods: A cross-sectional survey of 2270 adults was conducted in Singapore. Participants reported their religious affiliation to Buddhism, Christianity, Hinduism, Islam, Sikhism, Taoism, or other religions. A PMH instrument measured total PMH and six subcomponents: general coping (GC), emotional support (ES), spirituality (S), interpersonal skills (IS), personal growth and autonomy (PGA), and global affect (GA). Lifetime history of mental disorders was assessed with the Composite International Diagnostic Interview. Results: Total PMH (mean ± SD) was 4.56 ± 0.66 for participants with any religion versus 4.12 ± 0.63 (*p* = 0.002) in those without any religion. After adjustment for all potential confounders, the mean difference in total PMH between these groups was 0.348 (95% CI: 0.248–0.448). Having any religion was significantly associated with higher scores for S, GC, ES, IS, but not with PGA, GA or mental disorders. Compared with individuals without any religion, total PMH and S levels were significantly higher across all religions. Additionally, Christianity was significantly associated with higher ES, Taoism with higher GC, Buddhism and Islam with higher GC, ES and IS, Hinduism with higher IS and Sikhism with higher ES and IS. Conclusion: Our results indicate that religious affiliation is significantly associated with higher PMH, but not with mental disorders in an Asian community setting. In addition, different religions showed unique patterns of association with PMH subcomponents.

## 1. Introduction

Religion is an organized set of beliefs and practices that are followed individually or within a community, and often involve worship of a higher controlling power such as a personal God, gods or spirits [1]. It is also associated with spiritual practices that relate to feelings of closeness to the higher power, “self-transcendence and/or as engagement in practices” such as prayer or meditation [2]. Given its deep connection with values that individuals apply in their daily life, it is natural that religion and spirituality are considered influential social institutions having multifaceted relationships with aspects of people’s lives, including their mental health and health-related lifestyle habits such as diet, smoking and alcohol consumption [1]. As a result, therapies grounded in spiritual practices, also referred to as mind–body therapies such as mindfulness, have become popular in a number of countries, with some reporting that almost 20% of the US adult population practices some form of mind-body therapy [3]. Mindfulness-based cognitive behavioral therapy and breathing and relaxation techniques have also been widely adopted in clinical practice [4,5]. Although spiritual practices have gained clinical acceptance, mental health professionals have steered clear of religion-based applications such as prayer as a therapy [6]. This has been partly attributed to Sigmund Freud’s association of religion with neuroticism, and partly to the inconsistency of the evidence on the relation between religion and mental health [1]. 

Religious affiliation and religiosity can be linked to a person’s sense of hope, coping and social support networks [7,8], and has been associated with improved health and quality of life and reduced depression and anxiety in users of mental health services and those with serious medical conditions like cancers and cardiovascular diseases [9,10,11]. However, along with potentially beneficial effects on mental health, there are also reports of adverse experiences and functional outcomes such as prejudice, isolation, loss of autonomy, and poor logical thinking and analytical reasoning [12,13,14]. Negative religious coping and religiosity have also been associated with higher depression, stress and unhealthy lifestyle habits such as unhealthy diets, smoking and alcohol consumption [7,15,16]. Likewise, cultures in which prayer is highly valued have shown tendencies to shun psychiatric care thus hampering access to appropriate mental healthcare [17]. Hence, the role of religion in mental health is still being debated.

In relation to positive aspects of mental health and wellbeing, research pertaining to religious affiliation is limited to positive psychology and components of subjective wellbeing such as happiness and life satisfaction [18,19]. Association of religion with aspects of psychological well-being such as meaning in life and personal growth have been largely studied in Western societies [20]. This presents a major knowledge gap as distinct religions that are less presented in Western populations are believed to impact mental health differently due to the varying experiences, appraisal and values placed on positive emotions in individual religions which can be additionally influenced by prevailing religious norms [19,21]. 

This area of research has gained further interest by the fluctuating trends in the importance given to religion across the world. A 2010 global survey showed marked differences in the proportion of people identifying themselves as being religious in different geographic regions and population sub-groups [22]. The survey also indicated that younger adults aged 18 to 39 years were less likely to attribute importance to religion. While lower religiosity has been linked to high substance use, suicidality, and poor mental health in general among adolescents and young adults [23].

Singapore is an urban Southeast Asian country with a multi-ethnic resident (citizens and permanent residents) population of 4.03 million people comprising 74.4% Chinese, 13.4% Malay, 9% Indian and 3.2% other ethnic groups [24]. The religious composition in Singapore represents all major religions in Asia comprising Buddhism/Taoism (43.2%), Christianity (18.8%), Islam (14.0%), Hinduism (5%) and other religions such as Sikhism (0.6%). In addition, 18.5% of the Singapore population aged over 15 years old identify themselves as having no religion or being free thinkers [25]. Two previous studies in Singapore have reported differences in suicidality and use of mental health services by religious affiliation [26,27]. In another study, the association between intrinsic and extrinsic religiosity with happiness differed among adults in the United States and Singapore depending on their religious perceptions [28]. A more recent study observed that about 2.9% of Singaporean adults had used prayer or spiritual healing for their mental health problems in one year [29]. These studies indicate likely helpful and harmful associations between religious affiliation and mental health status in the population. Therefore, it was of interest to explore this association further with both positive and negative aspects of mental health. 

This cross-sectional study in a nationally representative sample of adults in Singapore aimed to:(i)Investigate whether religious affiliation to any religion or specific religions is associated with positive mental health and mental disorders; and(ii)Identify subcomponents of positive mental health that are associated with religious affiliations.

## 2. Materials and Methods

The present study used a subset of data from the Singapore Mental Health Study 2016, which was a national cross-sectional household survey conducted between August 2016 and April 2018. This was a survey on the prevalence of mental disorders as well as the assessment of positive mental health in Singapore’s adult population. Ethical approval was obtained from the National Healthcare Group’s Domain Specific Review Board prior to the start of the study. All respondents gave their written informed consent and consent was also obtained from parents or legal guardians of those aged below 21 years. The study particulars are published in a previous article [30].

Briefly, a disproportionate stratified sample was drawn from a national database of all Singapore residents where minority ethnic groups (Malays and Indians) and older age groups (65 years and above) were oversampled compared with the population proportions. A face-to-face household survey was conducted with Singapore Residents (citizens and permanent residents) aged 18 years and over, residing in Singapore at the time of the survey and who were able to understand the survey in the English, Chinese or Malay language. Residents who were institutionalized, uncontactable during the survey period or were unable to be interviewed due to severe physical or mental health problems or language ineligibility were excluded from the survey. 

A total of 6126 residents participated in the survey (response rate 69.5%). Of these, 4916 respondents who were literate in English were invited to complete a self-administered positive mental health instrument, which was available in only English language at the time of the survey. They were provided the questionnaire in a postage paid sealable envelope and were asked to mail the completed questionnaire back to the study team. This approach was used to allow privacy, provide adequate time to fill out the questionnaire, and minimize social desirability bias for the respondents. A total of 2337 respondents (47% of eligible participants) returned the completed questionnaires. Of these, 67 questionnaires were excluded due to pattern answers or missing data. Data from the remaining 2270 respondents was used in the current analysis. 

All respondents were offered interviewer-administered survey questionnaires that included a sociodemographic section and questions about lifestyle, height and weight, and history of physical and mental health conditions. 

### 2.1. Assessment of Religious Affiliation

Respondents were asked to indicate their religious affiliation from a list of predominant religions in Singapore. These were Buddhism, Christianity, Hinduism, Islam, Sikhism, Taoism, or other religions that were specified as open text in their responses. They were also given an option to denote if they were free-thinkers or had no religion (i.e., did not associate with any religion). Religious affiliation was also categorized into a dichotomous variable (yes or no), with all respondents who selected any religion grouped as “yes” and the rest classified as “no”.

### 2.2. Assessment of Positive Mental Health and Mental Disorders

#### 2.2.1. Positive Mental Health

The positive mental health (PMH) instrument is a 47-item self-administered instrument [31]. It has six subscales: general coping, emotional support, spirituality, interpersonal skills, personal growth and autonomy, and global affect. For the first five subscales, respondents are asked how much the items describe them on a scale from “not at all like me” to “exactly like me”. The spirituality subscale has items like, “I find comfort in my religion or spiritual beliefs,” “I set aside time for meditation or prayer,” and “I gain spiritual strength by trusting in a higher power.” The global affect subscale includes a list of five affect indicators and requires users to indicate how often over the past four weeks they felt calm, happy, peaceful, relaxed and enthusiastic, using a five-point response scale from “never or very rarely” to “very often or always.” Cronbach’s alpha for the instrument was found to be 0.951 in the study sample. All items have a weight of 1; the total PMH score was obtained by adding scores of all items and dividing by 47. Items belonging to the respective PMH subcomponents were summed and divided by the number of items in each subcomponent. Total and subcomponent scores range from 1 to 6, with higher scores indicating better PMH. In order to account for possible confounding of the relationship between PMH and religion by the spirituality subcomponent, for the current analysis, total PMH scores were also obtained after excluding the spirituality score.

#### 2.2.2. Mental Disorders

The Composite International Diagnostic Interview version 3.0 (CIDI 3.0) was used to assess lifetime diagnosis of major depressive disorder, bipolar disorder, generalized anxiety disorder, obsessive compulsive disorder and alcohol use disorders (alcohol abuse and/or dependence) using the Diagnostic and Statistical Manual of Mental Disorders, 4th Edition (DSM-IV) diagnostic algorithms with hierarchy rules [32]. Due to small sample size with specific mental disorders across religious affiliations, individual disorders were not investigated in greater detail in the current analysis, instead these were collectively used to classify presence of any mental disorder as yes or no. 

### 2.3. Assessment of Covariates

The survey asked about sociodemographic variables including age, gender, ethnicity, marital status, education level, and employment. Other covariates were body mass index, smoking status, and chronic physical conditions. Height (in meters) and weight (in kilograms) was self-reported by the respondents and used to calculate their body mass index (BMI) in kg/m^2^. Participants were also asked about their smoking status at the time of the survey as current smoker, ex-smoker or never smoked. The presence of any chronic physical condition was based on a self-report of ever being diagnosed by a doctor as having any of the following chronic physical conditions: respiratory disorders such as asthma or other chronic lung disease (chronic bronchitis or emphysema), diabetes, hypertension, hyperlipidemia, chronic pain (migraine, arthritis or rheumatism, back or spinal problems), cancers, neurological disorders (epilepsy, convulsion, Parkinson’s disease), cardiovascular disorders (stroke, coronary heart disease, angina, congestive heart failure or other heart disease), and ulcers and chronic inflamed bowel disorders. 

### 2.4. Statistical Analysis

Analyses were adjusted using sampling, non-response and stratification weights [30]. Means and standard deviation (SD) for the continuous variables and frequency distribution and standard error (SE) for categorical variables were computed to obtain the characteristics of the overall sample and by religious affiliation (yes or no-as the reference group). Bivariate analyses were conducted to assess the association of any religious affiliations with sociodemographic characteristics, mental disorders (individually and any mental disorder), and any chronic physical condition using chi-square tests. Possible multi-collinearity for ethnicity and religious affiliations was assessed using the variance inflation factor (VIF) and found to be below the acceptable level of less than 5. One-way ANOVA with Bonferroni corrections were used to test the differences in total PMH and PMH subcomponents scores across the different religious affiliations. The association between any and specific religious affiliations (independent variable) with total PMH, total PMH without spirituality score and PMH subcomponents as dependent variables were investigated with general linear regression models. Logistic regression models with Wald’s chi-square tests were used to assess association between any religious affiliation and any mental disorder (dependent variable). Five respondents belonging to other religions were not included in the multivariable analyses on specific religious affiliations due to small numbers. For the regression analysis, three models were tested. Model 1 assessed unadjusted association between the independent (religious affiliation) and dependent variables (PMH and any mental disorder). Model 2 was adjusted for basic sociodemographic covariates: age group (18–34, 35–49, 50–64, 65 and above), gender (women, men) and ethnicity (Chinese, Malay, Indian, Others). Besides these variables, Model 3 also included other potential confounders such as marital status (single, married, separated/divorced, widowed), education level (primary and below, secondary, junior college, vocational, diploma, university), employment (unemployed, economically inactive, employed), any lifetime mental disorder (yes/no; only for PMH models), history of any chronic physical condition (yes/no), BMI (kg/m^2^), and smoking status (current-smoker, ex-smoker, or never smoked). Beta estimates for differences in scores and odds ratios (OR) for differences in proportions were investigated with 95% confidence intervals (CIs). Two-sided statistical significance was set at *p* < 0.05. StataCorp, USA’s Stata SE 15 and International Business Machines (IBM) Corporation, New York, USA’s SPSS 24.0 Complex Samples were used for the analyses.

## 3. Results

Table 1 presents socio-demographic and health characteristics of the sample. There were 2270 respondents in the study, having a mean (SD) age of 42.1 (15.2) years, with slightly more women (52.1%) than men (47.9%). Of the participants, 29.8% followed Buddhism (*n* = 303), 25.3% Christianity (*n* = 475), 13.4% Islam (*n* = 798), 4.9% Taoism (*n* = 44), 4.3% Hinduism (*n* = 378), 0.4% Sikhism (*n* = 35), and rest followed other religions such as Jewish, New Age, or Parsee (*n* = 5), resulting in 78.2% with any religious affiliation. The remaining 21.8% reported having no religious affiliation. While there were no marked differences in the characteristics of the sample with and without a religious affiliation, age, ethnicity and marital status significantly varied between the groups (Table 1). Those affiliated to any religion were slightly older, more likely to be of non-Chinese ethnicity and married compared with those without a religious affiliation. Frequency of having a specific or any mental disorder and any chronic physical condition did not differ significantly in the two groups.

Total PMH, total PMH without spirituality score and PMH subcomponent scores, except for personal growth and autonomy and global affect, were higher in respondents affiliated to any religion than those not affiliated (Table 2). Although the estimates were lower, total PMH without spirituality score was higher among those with a religious affiliation. Almost all the associations, except for the global affect subcomponent, remained significant after controlling for basic demographic characteristics in Model 2 and all confounders in Model 3. For example, mean ± SD for total PMH in the sample affiliated to any religion was 4.56 ± 0.66 compared with 4.12 ± 0.63 in those without any religion. The mean total PMH level was estimated to be 0.437 higher in the population affiliated to any religion versus those without (*p* < 0.001). Accounting for the effect of all confounders, this mean difference was estimated to be 0.348 (95% CI: 0.248–0.448) in the population. As expected, the strongest association was observed for spirituality with a mean difference of 1.637 (95% CI: 1.428–1.845) after accounting for all potential confounders but pronounced associations with emotional support (β: 0.231; 95% CI:0.087–0.375) and general coping (β: 0.189; 95% CI: 0.052–0.326) were also observed. The estimates in the three models did not suggest any major confounding by sociodemographic or lifestyle characteristics or mental and physical health status in the association between religious affiliation and PMH (Table 2). 

Table 3 presents the differences in the PMH score and subcomponent between different religions affiliations. Those with no religious affiliation had significantly lower total PMH, emotional support and spirituality scores across all the religions. However, total PMH score without the spirituality showed a different pattern in the score distributions. For example, mean scores among Christians were not significantly higher than Buddhists. Besides these, several differences were observed between the religious groups across the six PMH subcomponents.

Results comparing PMH of the different religious groups with those without any religion are presented in Table 4. All religions (i.e., Christianity, Islam, Taoism, Buddhism, Hinduism, and Sikhism) were associated with higher total PMH and spirituality levels than those without religious affiliations in the three models. In contrast, personal growth and autonomy and global affect did not differ significantly compared to those without any religion. Additionally, after adjustment for all potential confounders, Christianity was associated with higher emotional support, Taoism with higher general coping, Buddhism and Islam with higher general coping, emotional support and interpersonal skills, Hinduism with higher interpersonal skills and Sikhism with higher emotional support and interpersonal skills. Religious affiliation was not significantly associated with having a history of any mental disorder. The odds ratio of having any mental disorder was 1.241 (95% CI: 0.899—1.654; *p* = 352) in bivariate analysis (Model 1), 1.385 (95% CI: 0.856—2.243, *p* = 0.185) after adjustment for basic sociodemographic variables (Model 2), and 1.266 (95% CI: 0.732—2.051; *p* = 0.337) after adjustment for all potential confounders (Model 3). Analysis with total PMH without spirituality scores as the dependent variable demonstrated lower strength of associations between PMH and specific religions with reference to no religious affiliation. This association was not observed for Christianity, Taoism and Hinduism (Models 2 and 3). Likewise, specific religious affiliations were not significantly associated with having any mental disorders (results not presented).

## 4. Discussion

This large study in a multi-ethnic Asian population contributes to literature on the association of religious affiliation with mental health by focusing on two complimentary aspects of mental health, PMH and mental disorders. First, we established that people identifying with any religious affiliation have better PMH. Second, we explored its association with PMH subcomponents. Third, we evaluated these associations independently for six predominant religions in Singapore, Christianity, Islam, Buddhism, Taoism, Hinduism, and Sikhism. Finally, we assessed whether having a religious affiliation was associated with having any mental disorder. Our results indicate that generally having a religious affiliation is significantly associated with higher PMH but not with mental disorders, and that different religions show unique patterns of association with PMH subcomponents, specifically emotional support and general coping. We discuss these findings in the context of previous studies and possible limitations.

Our results on PMH are supported by the limited literature available in the area of mental wellbeing, which indicates a direct association between religious affiliation and subjective and psychological wellbeing components such as happiness and coping in young, adults and elderly populations [8,21,33]. A study that investigated the association of religion with emotions among Christian, Muslim, Hindu, Buddhist and Jewish participants from 49 countries found that religion, “plays a role in the experience of pleasant (for example, love and gratitude) and unpleasant (such as guilt, shame, etc.) emotions,” and how desirable these emotions are to people within these religions [19], which in turn can influence their PMH. Cohen [34] investigated influence of different religions on mental wellbeing and proposed that variations in feelings of happiness among Christian and Jewish populations were due to differences in religious coping and beliefs. Similarly, Tsai et al. [35] found differences between Christians and Buddhists in relation to positive emotions. In this study, all examined religions were associated with higher total PMH as compared with those without any religious affiliation and we did not observe marked variations between the different religions. However, an important consideration in drawing comparisons with previous studies are that total PMH was measured by the PMH instrument that mainly represents psychological wellbeing and that our population sample was predominantly of Asian origin. These differences in the population and operational definition of PMH could explain the contrasts of our findings with the earlier research. 

A novel contribution of our study is the evaluation of the association of any and specific religious affiliations with six subcomponents of PMH. There is limited research on mental health for Eastern religions and settings. Hence, we are unable to conduct deeper comparisons with extant knowledge, however we provide some possible explanations for our observations. The association between religious affiliation and spirituality was naturally expected and believed to be brought about by social influences such as family, friends, pastors and teachers which is consistently reported in research studies conducted in Western, Mediterranean and Asian societies [36]. We also found that religious affiliation was associated with higher general coping, emotional support, and interpersonal skills. Coping, and particularly religious coping, has been widely studied in the past. A review of literature suggests there are direct and strong associations between religion and coping in general populations and in populations with specific health conditions such as cancers, cardiovascular diseases, schizophrenia, and depression [37]. The review concludes that people with illnesses find hope and meaning in life by leaning on their religious beliefs and practices to relieve their stress. Our study found that followers of Taoism, Buddhism and Islam had better general coping as compared to those without a religious affiliation. However, we did not observe this association with other religions such as Christianity or Hinduism. A recent review comparing coping and coping styles between Christian and non-Christian groups did not find substantial differences, however, they reported that styles of religious coping—whether they are adaptive or maladaptive and intrinsic or extrinsic—determine these associations which are stronger among non-Christian groups, likely due to their more frequent use of positive religious coping mechanisms [38]. An important finding from our study was that after removing the contribution of the spirituality subcomponent to total PMH scores, the strength of the associations lowered and populations following Christianity, Taoism, and Hinduism did not show higher total PMH versus those without a religious affiliation. This indicates a likely greater relevance of spirituality and religiosity to PMH of these groups. Further research is warranted to gain in-depth understanding into how spirituality/religiosity might differentially influence PMH in the population. Results should also be replicated in other populations and cultures to establish the association between religion and PMH.

In relation to emotional support and interpersonal skills, it has been proposed that community and social support built through active religious participation yield higher levels of these PMH subcomponents. Buddhism, Islam and Sikhism were significantly associated with better emotional support and interpersonal skills in our study. The role of emotional support is believed to be that of a mediator in the relationship between religion and mental distress and has been found to be the strong predictor of decreased hopelessness, depression, and suicide behaviors [39]. A study conducted in the USA found differences in emotional support between Catholics and non-Catholics based on whether their religious support profiles were secular, broad or limited [40]. Similarly, greater interpersonal skills were found to be associated with practicing religious traditions and church attendance [41]. In our study conducted in an Asian setting, Christianity was significantly associated with higher emotional support but not interpersonal skills, which could be due to prevailing religious practices in our population. Further research should thus look beyond religious affiliation to include religious practices as well as variations within these religions, for example those that may exist between different Christian groups or Zen traditions within Buddhism [19]. 

Of the six PMH subcomponents, the global affect did not show a significant association with any religion after accounting for the effect of other variables. This was unexpected. The global affect domain covers affective states such as being calm and peaceful, that have been widely linked to religiousness, especially in the case of Buddhism [42]. However, this relationship is stronger among older adults; age shows a curvilinear relationship with both religious affiliation and affective mental states [43]. Given that the average age of our sample was 42, it is possible that the associations between religious affiliations and global affect were weaker in our cohort and did not meet statistical significance, which could have resulted in the observed findings of our study. We also did not find any suggestion of a beneficial effect of religion on personal growth and autonomy. However, this finding is not entirely unexpected. Religious affiliation and religiousness have been associated with lower sense of control that is linked to autonomy [44]. In terms of personal growth, differences were previously observed between Christian and non-Christian students in the UK, depending on their religious participation, with higher participation related to lower personal growth [20]. It is proposed that the relation between religion and personal growth and autonomy is influenced by social and religious norms and whether the proponents of the respective religions perceive God as a secure base who can meet their needs for autonomy, competence and relatedness [45]. 

Our study thus uncovers the association of religion with not only overall PMH, but also provides insight into its contributing subcomponents and thus helps understand the mechanisms through which different religions may exert their positive influence on mental health.

While our study provides robust evidence on people with religious affiliation having higher PMH, our results do not suggest an inverse association with mental disorders. In contrast, we found somewhat higher odds of having any mental disorder, although these associations were not statistically significant. Studies conducted in Asian and Western populations have reported a higher prevalence of mental disorders in religious individuals compared to non-religious individuals [46,47]. However, various factors relating to religion and religiosity such as prayer, forgiveness, social support, and practice of religion, were found to be inversely associated with depression, anxiety, and lifetime risk of mental disorders consistently in various populations [48,49,50,51]. Although research findings in this area remain divided, certain explanations have been proposed to account for these contradictions. First, previous research in persons with mental disorders suggests that people with higher levels of mental distress may turn to spirituality and religion to cope with their situation and seek meaning and purpose [48,52]. Second, different mental disorders are believed to have dissimilar trajectories between religious affiliation and mental disorders. For example, having a religious affiliation was inversely associated with suicidality and alcohol abuse [46,48], not significantly associated with anxiety disorder, and directly associated with current depression [47]. Third, opposing associations have been observed between past and current depression [46] and between milder and severe symptoms of depression [53]. And lastly, religious norms that determine the perception of God and religiousness are believed to modify the association between religion and mental disorders [1,8]. For example, religions that perceive God as forgiving, tend to demonstrate an inverse association with mental disorders while those where God is linked to retribution, have been associated with a higher prevalence of mental disorders in a report investigating these associations in multiple religions [19]. However, due to the small numbers of participants with individual mental disorders and the lack of detailed information on religiosity, we were unable to investigate these issues in greater depth in our study.

There are important public health and policy implications of our study. We identified a direct association between religious affiliation and PMH with better PMH in four subcomponents: general coping, emotional support, interpersonal skills, and spirituality among those identifying with any religion. Consequently, they also had higher total PMH. Given the likely low overall PMH among those without any religious affiliation, it is important to target mental health promotion interventions towards this group. Specifically, addressing their coping and interpersonal skills and access to emotional support would be useful in enhancing their PMH. Measures that may improve individuals’ social capital for example through community participation or volunteering may be useful [54]. In addition, involving religious organizations and places of worship in mental health promotion would be beneficial at the population level. In a previous study in Singapore, 7.6% of the people with mental health problems had sought help from religious and spiritual organizations [55]. Our study indicates that religious and spiritual advisors are not only an important resource for people with mental disorders but they can also play a significant role in improving PMH in the population.

Our study addresses an important limitation of previous research that was highlighted by Kim-Prieto and Diener [19] who brought attention to the fact that most of the work in the past centered on Christianity and that it is important that findings are replicated for other religions including Buddhism, Islam, and Hinduism. Strengths of this study include the use of a large multi-ethnic and nationally representative sample, use of validated measures to assess PMH and mental disorders, and adjustment for potential confounders such as gender, smoking, chronic medical conditions, and BMI in the analysis [16,56]. However, certain limitations of our study need to be considered. First, in this study, only English-literate respondents who could self-administer the PMH instrument were included. Those who were not literate in English and therefore possibly older and with lower education were excluded which might have reduced the representativeness of the study. Past research indicates that these groups have greater religiosity [22], higher PMH [57], and a lower prevalence of mental disorders [30]. Therefore, this exclusion may have led to selection bias that would probably have resulted in estimates moving towards null, rather than strengthening the associations. Second, data on PMH and symptoms of mental disorders were obtained through self-report. Some level of social desirability bias is possible, specifically while answering items on spirituality, and this may have been more pronounced if some of these behaviors were discouraged in their affiliated religion. Peres et al. [33] suggested that individuals belonging to certain religious groups believe strongly in the “goodness of God and the need for gratitude to Him.” They may tend to over-report their positive emotion and under-report distress due to their perception that recognizing, “negative experiences would mean weakness or absence of their God or faith” [33]. Third, as our study was cross-sectional, it cannot conclusively distinguish the direction of cause and effect. Studies with a more detailed assessment of religious practices are needed to better understand the association of religion with mental health and whether a dose-response relationship exists with respect to the level of religiousness. Future research should also account for the effects of religious practices and attitudes while establishing the relationship between religion and PMH.

## 5. Conclusions

Our study addressed several limitations of past research and concurrently assessed the association of religious affiliation with PMH and mental disorder. To the best of our knowledge, this is the first study to have evaluated the link between multiple religions and mental health in a multi-ethnic Asian setting. We established that having a religious affiliation is associated with higher scores for several components of PMH including general coping and emotional support. In contrast, our findings do not suggest that having a religion is associated with a lower likelihood of having a mental disorder. However, further studies on religion and mental disorders are needed to replicate our findings in a clinical setting to ensure adequate power for detecting differences by individual mental disorders. Moreover, to assess the clinical relevance of this area of mental health research, prospective cohort studies are needed to evaluate the effects of religion on changes in mental health outcomes over time. This may enable planning of focused mental health promotion and possibly faith-based interventions in specific religious groups. Mental health services based in or delivered via places of religious worship such as churches or mosques may be effective in reducing anxiety and depression based on findings from observational studies and clinical trials [58,59]. Thus, taken together with the results of our study, there appears to be a strong rationale for pursuing more public health research on the relationship of religion with mental health outcomes.

## Figures and Tables

**Table 1 ijerph-18-03368-t001:** Socio-demographic characteristics of the sample and respondents with and without any religious affiliation.

		Overall Sample			Any Religious Affiliation			No Religious Affiliation			*p* Values #
Age (Mean, SD)		42.1, 15.2			42.9, 15.0			39.5, 15.4			0.002
BMI (Mean, SD)		24.4, 4.7			24.7, 4.8			23.1, 3.7			<0.001
		n	Wt. %	SE	n	Wt. %	SE	n	Wt. %	SE	
Age group	18–34 yrs.	679	34.9	0.1	576	32.2	0.8	103	44.4	3.0	0.008
	35–49 yrs.	618	33.3	0.0	564	34.5	0.8	54	29.3	2.9	
	50–64 yrs.	611	24.5	0.1	560	25.6	0.7	51	20.6	2.5	
	65+ yrs.	362	7.2	0.0	338	7.6	0.3	24	5.7	1.2	
Gender	Women	1162	52.1	1.5	1051	52.3	1.7	111	51.6	3.6	0.865
	Men	1108	47.9	1.5	987	47.7	1.7	121	48.4	3.6	
Ethnic group	Malay	603	10.4	0.1	602	13.2	0.3	5	0.1	0.1	<0.001
	Indian	693	8.1	0.1	682	10.2	0.2	11	0.6	0.2	
	Others	273	4.0	0.1	244	4.7	0.1	29	1.7	0.3	
	Chinese	701	77.5	0.2	510	71.9	0.5	191	97.6	0.4	
Marital status	Single	666	37.9	1.1	557	35.0	1.3	109	48.2	3.4	0.004
	Separated/divorced	106	4.9	0.7	95	4.8	0.7	11	5.2	1.6	
	Widowed	79	1.5	0.3	77	1.7	0.3	2	0.6	0.5	
	Married	1419	55.7	1.3	1309	58.4	1.5	190	46.1	3.4	
Education	Primary and below	176	4.6	0.6	168	4.8	0.6	8	3.6	1.3	0.064
	Secondary	590	20.2	1.1	547	20.5	1.2	43	19.0	2.7	
	Pre-U/Junior College	154	7.5	0.8	135	7.3	0.9	19	7.9	1.9	
	Vocational Institute/ITE	160	5.5	0.6	156	6.4	0.7	4	2.1	1.0	
	Diploma	461	22.4	1.3	414	23.0	1.4	47	20.2	2.9	
	University	728	39.9	1.4	617	37.9	1.6	111	47.2	3.5	
Employment status	Unemployed	114	5.1	0.7	100	4.6	0.7	14	6.8	1.8	0.404
	Economically inactive	585	20.6	1.1	531	20.5	1.2	54	21.0	2.8	
	Employed	1571	74.3	1.2	1407	74.9	1.3	164	72.1	3.1	
Smoking status	Ex-smoker	261	9.7	0.9	240	82.6	3.9	21	17.4	3.9	0.082
	Never smoked	1634	77.6	1.2	1447	76.6	1.6	187	23.4	1.6	
	Current smoker	373	12.8	0.9	349	84.1	3.3	24	15.9	3.3	
Any mental disorder ^	Yes	319	14.8	1.1	284	15.4	1.2	35	12.8	2.4	0.352
	No	1951	85.2	1.1	1754	84.6	1.2	197	87.2	2.4	
Major depressive disorder	Yes	131	6.6	0.8	117	6.9	0.9	14	5.5	1.6	0.456
	No	2139	93.4	0.8	1921	93.1	0.9	218	94.5	1.6	
Bipolar disorder	Yes	32	1.8	0.4	26	1.6	0.4	6	2.7	1.2	0.270
	No	2238	98.2	0.4	2012	98.4	0.4	226	97.3	1.2	
Generalized anxiety disorder	Yes	47	2.1	0.4	41	2.2	0.5	6	1.7	0.9	0.648
	No	2223	97.9	0.4	1997	97.8	0.5	226	98.3	0.9	
Obsessive compulsive disorder	Yes	92	4.4	0.6	83	4.3	0.7	9	4.8	1.6	0.773
	No	2178	95.6	0.6	1955	95.7	0.7	223	95.2	1.6	
Alcohol use disorder	Yes	90	4.6	0.6	79	4.8	0.7	11	3.9	1.4	0.594
	No	2180	95.4	0.6	1959	95.2	0.7	221	96.1	1.4	
Any chronic physical condition ^^	Yes	1301	51.8	1.5	1185	52.6	1.6	116	48.8	3.5	0.334
	No	967	48.2	1.5	851	47.4	1.6	116	51.2	3.5	

Wt.: weighted estimates adjusted for sampling, post-stratification and non-response weights; SE: Stan; # Generalized linear model for continuous and chi-square tests for categorical variables. ^ Any mental disorder refers to having at least one of the conditions assessed in this study (i.e., major depressive, bipolar, generalized anxiety, obsessive compulsive, and/or alcohol use disorder). ^^ Any chronic physical condition refers to self-report of at least one diagnosed physical health condition (i.e., respiratory disorders and other chronic lung disease, diabetes, hypertension, hyperlipidemia, chronic pain, cancers, neurological disorders, cardiovascular disorders, ulcers and/or chronic inflamed bowel disorders).

**Table 2 ijerph-18-03368-t002:** Association between having any religious affiliation and total positive mental health and its subcomponents.

	Model 1				Model 2				Model 3			
	Mean	SD	β	*p*	β	95% CI		*p*	β	95% CI		*p*
						Lower	Upper			Lower	Upper	
Total positive mental health												
Any religious affiliation	4.56	0.66	0.437	<0.001	0.339	0.237	0.441	<0.001	0.348	0.248	0.448	<0.001
No religion	4.12	0.63			Ref				Ref			
Total positive mental health ^@^												
Any religious affiliation	4.08	0.59	0.155	<0.001	0.102	0.005	0.200	0.040	0.105	0.010	0.201	0.030
No religion	3.92	0.61			Ref				Ref			
General coping												
Any religious affiliation	4.56	0.84	0.244	<0.001	0.190	0.052	0.329	0.007	0.189	0.052	0.326	0.007
No religion	4.32	0.88			Ref				Ref			
Emotional support												
Any religious affiliation	4.80	0.87	0.269	<0.001	0.229	0.077	0.382	0.003	0.231	0.087	0.375	0.002
No religion	4.53	0.96			Ref				Ref			
Spirituality												
Any religious affiliation	3.96	1.51	1.930	<0.001	1.593	1.388	1.798	<0.001	1.637	1.428	1.845	<0.001
No religion	2.03	1.18			Ref				Ref			
Interpersonal skills												
Any religious affiliation	4.69	0.73	0.177	0.002	0.125	0.004	0.246	0.043	0.125	0.004	0.246	0.043
No religion	4.52	0.74			Ref				Ref			
Personal growth and autonomy												
Any religious affiliation	4.61	0.79	0.053	0.391	−0.035	−0.163	0.093	0.589	−0.022	−0.148	0.103	0.725
No religion	4.56	0.79			Ref				Ref			
Global affect												
Any religious affiliation	4.56	0.66	0.168	0.012	0.138	−0.002	0.278	0.053	0.136	−0.002	0.275	0.053
No religion	4.12	0.63			Ref				Ref			

β: Beta coefficient;**^@^**Total PMH without spirituality score. Model 1: unadjusted, bivariate general linear regression model; Model 2: general linear regression model, adjusted for age, gender, ethnicity; Model 3: general linear regression model, adjusted for age, gender, ethnicity, marital status, education level, employment status, smoking status, body mass index, any mental disorder and history of any chronic physical condition.

**Table 3 ijerph-18-03368-t003:** Differences in total positive mental health and subcomponent scores across different religions affiliations.

	Mean	SD	95% CI		Mean Difference in PMH Scores					
			Lower	Upper	Christianity	Taoism	Buddhism	Hinduism	Islam	Sikhism
Total positive mental health										
Christianity	4.61	0.68	4.53	4.69	0.00					
Taoism	4.36	0.57	4.20	4.53	−0.25	0.00				
Buddhism	4.41	0.61	4.33	4.48	−0.21 *	0.04	0.00			
Hinduism	4.66	0.68	4.59	4.73	−0.05	0.30	0.25 *	0.00		
Islam	4.80	0.66	4.75	4.85	0.19 *	0.44 *	0.39 *	0.14 ^#^	0.00	
Sikhism	4.77	0.52	4.61	4.94	0.16	0.41	0.36 ^#^	0.11	−0.03	0.00
No religion	4.12	0.63	4.03	4.20	−0.49 *	−0.25	−0.29 *	−0.54 *	−0.68 *	−0.65 *
Total positive mental health ^@^										
Christianity	4.02	0.60	3.95	4.09	0.00					
Taoism	4.02	0.54	3.86	4.18	0.00	0.00				
Buddhism	4.07	0.57	4.00	4.14	0.05	0.05	0.00			
Hinduism	4.14	0.61	4.07	4.20	0.12	0.12	0.07	0.00		
Islam	4.18	0.61	4.13	4.23	0.16 *	0.16	0.11	0.04	0.00	
Sikhism	4.26	0.47	4.11	4.41	0.24	0.24	0.19	0.12	0.08	0.00
No religion	3.92	0.61	3.84	4.00	−0.10	−0.10	−0.15	−0.22 *	−0.26 *	−0.34 ^#^
General coping										
Christianity	4.51	0.83	4.41	4.61	0.00					
Taoism	4.61	0.76	4.38	4.83	0.10	0.00				
Buddhism	4.56	0.78	4.47	4.65	0.05	−0.05	0.00			
Hinduism	4.47	0.99	4.37	4.58	−0.04	−0.13	−0.09	0.00		
Islam	4.68	0.92	4.61	4.75	0.17 ^	0.08	0.12	0.21 ^	0.00	
Sikhism	4.65	0.84	4.36	4.93	0.14	0.04	0.09	0.17	−0.04	0.00
No religion	4.32	0.88	4.20	4.44	−0.19	−0.29	−0.24	-0.15	−0.36 *	−0.33
Emotional support										
Christianity	4.77	0.89	4.66	4.88	0.00					
Taoism	4.65	0.95	4.38	4.93	−0.12	0.00				
Buddhism	4.79	0.89	4.69	4.88	0.02	0.14	0.00			
Hinduism	4.82	0.95	4.71	4.92	0.05	0.16	0.03	0.00		
Islam	4.94	0.88	4.88	5.01	0.17 ^#^	0.29	0.16	0.13	0.00	
Sikhism	5.11	0.68	4.90	5.33	0.34	0.46	0.33	0.30	0.17	0.00
No religion	4.53	0.96	4.40	4.67	−0.24 ^#^	−0.12	−0.25 ^#^	-0.28 ^	-0.41 *	-0.58 ^
Spirituality										
Christianity	4.68	1.28	4.52	4.83	0.00					
Taoism	2.91	1.30	2.53	3.29	−1.77 *	0.00				
Buddhism	3.02	1.29	2.87	3.17	−1.66 *	0.11	0.00			
Hinduism	4.31	1.31	4.17	4.45	−0.37 *	1.40 *	1.29 *	0.00		
Islam	4.96	1.00	4.88	5.04	0.28 ^	2.05 *	1.94 *	0.65 *	0.00	
Sikhism	4.30	1.49	3.75	4.84	−0.38	1.39 *	1.28 *	−0.01	−0.66 ^#^	0.00
No religion	2.03	1.18	1.87	2.19	−2.65 *	−0.88 *	−0.99 *	−2.28 *	−2.93 *	−2.27 *
Interpersonal skills										
Christianity	4.63	0.75	4.54	4.72	0.00					
Taoism	4.60	0.68	4.40	4.80	–0.03	0.00				
Buddhism	4.68	0.76	4.60	4.77	0.05	0.08	0.00			
Hinduism	4.81	0.71	4.74	4.89	0.18 ^	0.21	0.13	0.00		
Islam	4.82	0.74	4.75	4.88	0.19 *	0.21	0.13	0.01	0.00	
Sikhism	4.93	0.63	4.71	5.15	0.30	0.33	0.25	0.12	0.11	0.00
No religion	4.52	0.74	4.41	4.62	−0.11	−0.08	−0.16	−0.29 *	−0.30 *	–0.41 ^#^
Personal growth and autonomy										
Christianity	4.52	0.82	4.42	4.62	0.00					
Taoism	4.53	0.66	4.34	4.73	0.01	0.00				
Buddhism	4.61	0.71	4.52	4.70	0.09	0.08	0.00			
Hinduism	4.82	0.77	4.74	4.90	0.30 *	0.29	0.21 ^#^	0.00		
Islam	4.72	0.82	4.66	4.78	0.20 *	0.19	0.11	−0.11	0.00	
Sikhism	4.85	0.68	4.63	5.08	0.33	0.32	0.24	0.03	0.14	0.00
No religion	4.56	0.79	4.45	4.67	0.04	0.02	−0.06	−0.27 ^	−0.16	−0.30
Global affect										
Christianity	4.64	0.85	4.54	4.75	0.00					
Taoism	4.79	0.87	4.53	5.05	0.15	0.00				
Buddhism	4.64	0.82	4.54	4.74	0.01	−0.15	0.00			
Hinduism	4.68	0.90	4.58	4.77	0.03	−0.11	0.04	0.00		
Islam	4.72	0.88	4.65	4.79	0.07	−0.07	0.08	0.04	0.00	
Sikhism	4.73	0.66	4.49	4.97	0.09	−0.06	0.09	0.05	0.01	0.00
No religion	4.50	0.86	4.38	4.62	−0.14	−0.29	−0.14	−0.18	−0.22 ^#^	−0.23

^@^ Total PMH without spirituality score. One-way ANOVA with Bonferroni correction, two-sided *p* < 0.05 (^#^), *p* < 0.01 (^), and *p* < 0.001 (*).

**Table 4 ijerph-18-03368-t004:** Association between specific religious affiliations and positive mental health subcomponents.

	Model 1				Model 2				Model 3			
	β	95% CI		P	β	95% CI		P	β	95% CI		P
		Lower	Upper			Lower	Upper			Lower	Upper	
Total positive mental health												
Christianity	0.494	0.375	0.614	<0.001	0.454	0.330	0.579	<0.001	0.443	0.320	0.566	<0.001
Taoism	0.245	0.059	0.432	0.010	0.203	0.012	0.394	0.037	0.228	0.042	0.414	0.016
Buddhism	0.289	0.176	0.402	<0.001	0.275	0.162	0.388	<0.001	0.282	0.170	0.394	<0.001
Hinduism	0.543	0.430	0.656	<0.001	0.399	0.215	0.583	<0.001	0.361	0.182	0.540	<0.001
Islam	0.681	0.581	0.782	<0.001	0.608	0.413	0.803	<0.001	0.587	0.391	0.782	<0.001
Sikhism	0.653	0.466	0.839	<0.001	0.509	0.280	0.739	<0.001	0.477	0.253	0.700	<0.001
No religion	Ref				Ref				Ref			
Total positive mental health ^@^												
Christianity	0.099	−0.012	0.211	0.079	0.062	−0.052	0.177	0.287	0.058	−0.056	0.171	0.321
Taoism	0.098	−0.081	0.277	0.284	0.064	−0.12	0.247	0.498	0.079	−0.099	0.258	0.384
Buddhism	0.149	0.04	0.257	0.007	0.132	0.023	0.24	0.017	0.139	0.032	0.247	0.011
Hinduism	0.217	0.111	0.322	<0.001	0.104	−0.06	0.268	0.215	0.072	−0.089	0.233	0.379
Islam	0.261	0.165	0.357	<0.001	0.214	0.041	0.388	0.016	0.2	0.028	0.372	0.022
Sikhism	0.341	0.171	0.512	<0.001	0.23	0.024	0.436	0.029	0.215	0.008	0.422	0.042
No religion	Ref				Ref				Ref			
General coping												
Christianity	0.188	0.030	0.346	0.020	0.165	0.002	0.329	0.047	0.138	−0.026	0.302	0.099
Taoism	0.286	0.030	0.542	0.029	0.224	−0.041	0.488	0.097	0.258	0.013	0.502	0.039
Buddhism	0.238	0.085	0.391	0.002	0.213	0.062	0.365	0.006	0.209	0.058	0.360	0.007
Hinduism	0.153	−0.009	0.315	0.064	0.098	−0.160	0.357	0.455	0.034	−0.216	0.283	0.792
Islam	0.361	0.219	0.504	<0.001	0.404	0.121	0.686	0.005	0.339	0.063	0.615	0.016
Sikhism	0.327	0.019	0.634	0.037	0.29	−0.075	0.654	0.119	0.280	−0.054	0.614	0.101
No religion	Ref				Ref				Ref			
Emotional support												
Christianity	0.235	0.063	0.408	0.008	0.228	0.049	0.407	0.013	0.192	0.023	0.362	0.026
Taoism	0.117	−0.189	0.423	0.454	0.131	−0.170	0.432	0.392	0.139	−0.159	0.437	0.361
Buddhism	0.253	0.088	0.417	0.003	0.257	0.090	0.423	0.003	0.268	0.109	0.427	0.001
Hinduism	0.281	0.113	0.449	0.001	0.233	−0.007	0.472	0.057	0.178	−0.059	0.416	0.140
Islam	0.410	0.261	0.558	<0.001	0.375	0.141	0.609	0.002	0.361	0.118	0.605	0.004
Sikhism	0.579	0.326	0.832	<0.001	0.531	0.215	0.847	0.001	0.468	0.137	0.799	0.006
No religion	Ref				Ref				Ref			
Spirituality												
Christianity	2.648	2.423	2.873	<0.001	2.552	2.315	2.789	<0.001	2.575	2.345	2.806	<0.001
Taoism	0.880	0.464	1.295	<0.001	0.787	0.369	1.205	<0.001	0.869	0.451	1.287	<0.001
Buddhism	0.992	0.767	1.216	<0.001	0.964	0.744	1.185	<0.001	1.001	0.776	1.226	<0.001
Hinduism	2.282	2.069	2.495	<0.001	1.924	1.579	2.269	<0.001	1.949	1.604	2.295	<0.001
Islam	2.929	2.748	3.110	<0.001	2.647	2.279	3.015	<0.001	2.676	2.295	3.057	<0.001
Sikhism	2.268	1.699	2.837	<0.001	1.913	1.275	2.55	<0.001	1.888	1.244	2.532	<0.001
No religion	Ref				Ref				Ref			
Interpersonal skills												
Christianity	0.113	−0.026	0.251	0.110	0.097	−0.047	0.241	0.187	0.082	−0.063	0.226	0.268
Taoism	0.086	−0.141	0.312	0.459	0.066	−0.163	0.296	0.571	0.069	−0.159	0.297	0.552
Buddhism	0.165	0.031	0.298	0.016	0.163	0.029	0.297	0.017	0.154	0.018	0.29	0.026
Hinduism	0.294	0.166	0.422	<0.001	0.247	0.033	0.461	0.024	0.214	0.001	0.425	0.044
Islam	0.299	0.178	0.421	<0.001	0.348	0.111	0.585	0.004	0.311	0.075	0.546	0.010
Sikhism	0.410	0.169	0.652	0.001	0.336	0.04	0.632	0.026	0.340	0.051	0.63	0.021
No religion	Ref				Ref				Ref			
Personal growth and autonomy												
Christianity	−0.038	−0.187	0.111	0.621	−0.105	−0.260	0.05	0.185	−0.102	−0.252	0.048	0.182
Taoism	−0.025	−0.248	0.199	0.830	−0.091	−0.315	0.133	0.427	−0.074	−0.295	0.148	0.514
Buddhism	0.055	−0.088	0.199	0.448	0.029	−0.113	0.172	0.687	0.043	−0.097	0.184	0.545
Hinduism	0.267	0.130	0.404	<0.001	−0.022	−0.235	0.191	0.838	−0.057	−0.260	0.146	0.582
Islam	0.161	0.035	0.287	0.012	−0.034	−0.252	0.184	0.761	−0.038	−0.247	0.171	0.720
Sikhism	0.297	0.045	0.549	0.021	0.026	−0.269	0.322	0.862	−0.026	−0.318	0.266	0.861
No religion	Ref				Ref				Ref			
Global affect												
Christianity	0.145	−0.014	0.303	0.073	0.115	−0.049	0.279	0.168	0.100	−0.063	0.264	0.228
Taoism	0.290	0.004	0.577	0.047	0.283	−0.004	0.571	0.053	0.294	−0.006	0.594	0.055
Buddhism	0.139	−0.016	0.295	0.079	0.141	−0.015	0.297	0.077	0.134	−0.022	0.289	0.092
Hinduism	0.176	0.022	0.331	0.025	0.157	−0.112	0.426	0.254	0.087	−0.176	0.350	0.517
Islam	0.219	0.080	0.358	0.002	0.202	−0.106	0.510	0.199	0.169	−0.141	0.478	0.285
Sikhism	0.231	−0.037	0.500	0.091	0.187	−0.178	0.552	0.315	0.117	−0.249	0.484	0.530
No religion	Ref				Ref				Ref			

β: Beta coefficient; ^@^ Total PMH without spirituality dimension scores. Model 1: unadjusted, bivariate general linear regression model; Model 2: General linear regression model, adjusted for age, gender, ethnicity; Model 3: general linear regression model, adjusted for age, gender, ethnicity, marital status, education, employment status, smoking status, body mass index, any mental disorder, and history of any chronic physical condition.

## Data Availability

The data are not accessible to the public due to funding requirements. The data may be available from the corresponding author upon reasonable request.

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
