# Peer review of "Religious Affiliation in Relation to Positive Mental Health and Mental Disorders in a Multi-Ethnic Asian Population"

_ijerph, 2021, doi:10.3390/ijerph18073368_

Round 1

Reviewer 1 Report

He study presented aims to analyze how belonging to each of the religions included in the study mediate health variables and mental problems. The introduction is broad enough to give the reader an adequate background. The proposed method is adequate, although instruments could have been used to measure symptomatic groups that could be more faithful to the levels of mental health, screening the level of symptoms that the population presents according to whether or not they belong to any of the religions. However, the approach is adequate and they collect an important sample. The main limitation that must be corrected is that the number of subjects of each religion is not indicated in the sociodemographic table and therefore, we do not know how each of them are represented. This is information that must be reported. As well as the results are indicated adequately, the conclusions should highlight the conclusions, present the limitations of the study (replicate in other countries to control possible strange variables inserted in the culture of the country, for example) and propose future study proposals. Actions are proposed that require a more detailed analysis, such as places of worship being places where actions are carried out to promote mental health, this is not fully supported by the results. I hope I have clarified which are the minor problems that must be corrected, such as informing about the number of subjects corresponding to each religion to see the representation and if they are equated and the conclusions must be in accordance with the indications previously exposed. Sorry for my brevity in the previous report. Receive a cordial greeting.

Author Response

He study presented aims to analyze how belonging to each of the religions included in the study mediate health variables and mental problems. The introduction is broad enough to give the reader an adequate background. The proposed method is adequate, although instruments could have been used to measure symptomatic groups that could be more faithful to the levels of mental health, screening the level of symptoms that the population presents according to whether or not they belong to any of the religions. However, the approach is adequate and they collect an important sample.

Thank you for your encouraging comments. We take note of the comment on illness severity and will incorporate it into future investigations in this area.

The main limitation that must be corrected is that the number of subjects of each religion is not indicated in the sociodemographic table and therefore, we do not know how each of them are represented. This is information that must be reported.

>>Thanks for highlighting this. We have added the participant breakdown for each religion in the revised results section. Please note, the proportions quoted in the results are weight-adjusted.

As well as the results are indicated adequately, the conclusions should highlight the conclusions, present the limitations of the study (replicate in other countries to control possible strange variables inserted in the culture of the country, for example) and propose future study proposals. Actions are proposed that require a more detailed analysis, such as places of worship being places where actions are carried out to promote mental health, this is not fully supported by the results.

>>We agree with the limitations highlighted by the reviewer. We have added these to the study limitations and have also covered the role of places of worship in mental health promotion in study implications as advised by the second reviewer. We hope these address your comments.

I hope I have clarified which are the minor problems that must be corrected, such as informing about the number of subjects corresponding to each religion to see the representation and if they are equated and the conclusions must be in accordance with the indications previously exposed. Sorry for my brevity in the previous report. Receive a cordial greeting.

>>Thanks for summarising the key points. Our responses to these are as explained above.

Reviewer 2 Report

This cross-section survey using data from Singapore Mental Health Survey of 2016 examined the association between religious affiliation and positive mental health and mental disorders. The study added new information on the overall relationship between religion and positive mental health but also by looking at the subcomponents of positive mental health may suggest some possible mechanisms through which religions promote positive mental health. The following issues are raised about the manuscript:

  1. Only 37% of the original sample were respondents in the current analysis and a sub-sample of the original sample that were literate in English participated in the completing the Positive Mental Health Instrument. In the discussion, the authors indicated that they utilized a nationally representative sample but justification of this statement needed more explanation given the large sample loss and the focus on an exclusive sub-sample from the survey.
  2. Total Positive Mental Health included a subcomponent of “spirituality” and this subcomponent may be confounding the relationship between religious affiliations and the total Positive Mental Health. I would have suggested that the analysis be done including and excluding this subcomponent from the Total score.
  3. The discussion section should have discussed possible public health implications of the findings that religious affiliation was significantly associated with higher positive mental health.

Author Response

This cross-section survey using data from Singapore Mental Health Survey of 2016 examined the association between religious affiliation and positive mental health and mental disorders. The study added new information on the overall relationship between religion and positive mental health but also by looking at the subcomponents of positive mental health may suggest some possible mechanisms through which religions promote positive mental health. The following issues are raised about the manuscript:

1.Only 37% of the original sample were respondents in the current analysis and a sub-sample of the original sample that were literate in English participated in the completing the Positive Mental Health Instrument. In the discussion, the authors indicated that they utilized a nationally representative sample but justification of this statement needed more explanation given the large sample loss and the focus on an exclusive sub-sample from the survey.

 >>Thanks for pointing this. We agree with the reviewer’s emphasis on the representativeness of the study. We have tried to adjust for selection bias imposed by the inclusion criteria by adjusted the data with non-response weights. Nevertheless, we have removed the words nationally representative from the specified sentence in the discussion and also indicated the limitation in the manuscript. We hope the revisions are adequate.

2.Total Positive Mental Health included a subcomponent of “spirituality” and this subcomponent may be confounding the relationship between religious affiliations and the total Positive Mental Health. I would have suggested that the analysis be done including and excluding this subcomponent from the Total score.

>> Thanks for the suggestion. We have added this analysis in the article.

3.The discussion section should have discussed possible public health implications of the findings that religious affiliation was significantly associated with higher positive mental health.

>>Thank you for the helpful comment. We have added public health implications of our study to the revised discussion.